# Performing Advanced Trauma Life Support (ATLS) across Borders: Midterm Follow-Up of the Aeromedical Evacuation after Civilian Bus Accident at Madeira

**DOI:** 10.3390/jcm12144556

**Published:** 2023-07-08

**Authors:** Sebastian Imach, Andreas Deschler, Stefan Sammito, Miguel Reis, Sylta Michaelis, Beneditk Marche, Thomas Paffrath, Bertil Bouillon, Thorsten Tjardes

**Affiliations:** 1Department of Trauma and Orthopedic Surgery, Cologne-Merheim Medical Center (CMMC), University Witten/Herdecke, 51109 Cologne, Germanytjardest@kliniken-koeln.de (T.T.); 2Special Air Mission Wing, Federal Ministry of Defence, 51147 Cologne, Germany; 3Department of Anesthesiology, Intensive Care, Emergency Medicine and Pain Therapy, Bundeswehr Central Hospital, 56072 Koblenz, Germany; 4Experimental Aerospace Medicine Research, German Air Force Centre of Aerospace Medicine, 51147 Cologne, Germany; 5Department of Occupational Medicine, Medical Faculty, Otto von Guericke University, 39106 Magdeburg, Germany; 6Department of Surgery, Serviço Regional de Saúde da Madeira, 6180 Funchal, Portugal; 7Department of Trauma Surgery, Hospital of the Augustinerinnen, 50678 Cologne, Germany

**Keywords:** traumatology, trauma surgery, disaster medicine, patient transfer, trauma rehabilitation, Advanced Trauma Life Support (ATLS), trauma care, air ambulances

## Abstract

On 17 April 2019, a coach with tourists from Germany crashed in Madeira, requiring repatriation by the German Air Force. The Advanced Trauma Life Support (ATLS) concept was the central component of patient care. Data in Madeira were collected through a structured interview. The analysis of the Aeromedical Evacuation was based on intensive care transport records. In Germany, all available medical data sheets were reviewed for data collection. Quality of life (HRQoL) was evaluated by the 12-item Short Form Health Survey (SF-12). Twenty-eight prehospital patients were transported to the Level III Trauma Center in Funchal (Madeira). Five operative procedures were performed. Fifteen patients were eligible for Aeromedical Evacuation (AE). In the second hospital phase in Germany, in total 82 radiological images and 9 operations were performed. Hospital stay lasted 11 days (median, IQR 10–18). Median follow-up (14 of 15 patients) was 16 months (IQR 16–21). Eighty percent (8 out of 10) showed an increased risk for post-traumatic stress disorder (PTSD). Six key findings were identified in this study: divergent injury classification, impact of AE mission on health status, lack of communication, need of PTSD prophylaxis, patient identification, and media coverage. Those findings may improve AE missions in the future, e.g., when required after armed conflicts.

## 1. Introduction

On 17 April 2019, a coach (Irizar type) carrying a German civilian tour group crashed in Caniço in the Portuguese island of Madeira. The bus crashed 200 m after starting on a steeply sloping road on a left-hand bend, falling down a steep embankment. It overturned sideways and collided with the roof of a house before stopping (Figure 1). The roof of the bus was significantly deformed and partially ripped open. Only five passengers were found in the vehicle, the others were presumably ejected. Seat belts are compulsory in buses in Portugal. The cause of the accident is said to be excessive speed. A technical defect in the vehicle has been ruled out. Fatally injured were 28 of the 56 bus occupants. Subsequently, another patient died in hospital. All the deceased were German citizens. The injured bus driver and the tour guide were Portuguese. All injured patients were taken to the hospital Serviço Regional de Saúde da Região Autónoma da Madeira (SESARAM, EPRAM) in Funchal, the capital of Madeira, after receiving prehospital care.

The German Air Force was assigned to repatriating the surviving patients following a visit by Germany’s Foreign Minister, Heiko Mass, on 18 April 2019. The repatriation of 15 transferable patients was carried out airborne using the Aeromedical Evacuation (AE) capability in the German Air Force Airbus A310-304 Multi-Role Transport Tanker (MRTT) to Cologne Bonn Airport in Germany on 20 April 2019 [1]. This date was also a Germany-wide holiday as part of the Christian Easter celebration. For further inpatient care, the patients were transferred to the Cologne Merheim Hospital, a Level 1 trauma center 13 km away from the airport. The first inpatient admission was scheduled on 18:00 on the day of transfer.

This study describes the medical and logistical challenges in delivering ATLS-based management in a mass casualty event requiring repatriation and reports on quality of life outcomes two years after the accident.

## 2. Materials and Methods

### 2.1. Madeira

Prehospital deployment details (staff) and data from the initial clinical care (ATLS-compliance of care and injuries identified) in Madeira were collected through a structured interview with the medical directors of prehospital care and of the hospital. Also, the required staff was documented.

### 2.2. Description of the Aeromedical Evacuation German Air Force

The analysis of the AE was based on an evaluation of the intensive care transport records used during flight (based on recommendations of the German Interdisciplinary Association for Intensive Care and Emergency Medicine [DIVI], version 1.1, including all vital signs and medical treatments) [2]. These records were started by the attending intensive care physician after the patient was handed over by the emergency service at the airport of embarkation, until the patient was handed over to the emergency service at the airport of debarkation. Data were checked for plausibility and missing data were added by double-checking the paper-based data sheets (two cases). Potential channels of in-flight communication were critically appraised on data security and all-time availability. Staff requirements were documented.

### 2.3. Cologne

Patient charts, digital physician documentation, emergency department software, and the Madeira workgroup’s lessons learned protocol were reviewed for data collection. Information of medical assessment and management, additional injuries identified, and the staffing involved were extracted. 

The patients who were temporally and spatially oriented (10 of 15) were screened for psychological trauma sequelae on the day after admission using a standardized screening tool (Freiburg Screening Questionnaire (FSQ)) for Identification of Patients at Risk of Development of a Post-Traumatic Stress Disorder (PTSD) in order to define the need of talk therapy [3]. FSQ consists of 10 self-reported, dichotomous questions giving 1 point each. At a cut-off value of 3 or higher, FSQ is able to predict the risk of development of PTSD 6 months after a severe road accident needing surgical treatment with 87% sensitivity and 69% specificity [4].

### 2.4. Follow-Up

To assess health-related quality of life (HRQoL) and individual patient satisfaction, the patients were given a 12-item Short Form Health Survey (SF-12) and five additional questions evaluating their satisfaction with treatment, work capacity, and trauma-related medical treatment. The SF-12 is a self-reported, health-related quality of life questionnaire using 12 questions from 8 dimensions of health (physical health-related: general health, physical functioning, role-physical, and body pain; mental health related: vitality, social functioning, role-emotional, and mental health). For the evaluation of the SF-12, the corresponding German reference collective was used. SF-12 as a short version of the Medical Outcome Study 36-Item Short Form Survey (SF-32) reports two summary scores: the physical component score (PCS-12) and the mental component score (MCS-12). Local population average scores of PCS-12 and MCS-12 are both at 50 points. A 10-point deviation represents one standard deviation from the average (Z-score) [5]. Thus, the HRQoL assessment concept of the TraumaRegister DGU^®^ was implemented [6].

Patients were contacted by mail for the first time 15 months after the accident. A reminder was sent after 18 and 21 months. In case of ambiguity, telephone contact was established after the patient gave written consent.

### 2.5. Statistical Analysis

Statistical analysis was performed using SPSS statistical software (version 24; IBM Inc., Armonk, NY, USA). Data between groups were compared using the Mann–Whitney U test for continuous variables and Fisher’s exact test for categorical variables, unless indicated otherwise. 

A significance level of *p* < 0.05 was applied. Data were presented as medians with IQRs for continuous variables and as percentages for incidence rates.

## 3. Results

### 3.1. Patients

The median patient age was 69 years (interquartile range [IQR] 67–73), with nine female and six male patients. Ten (66%) had chronic pre-existing illnesses (four arterial hypertension; three coronary artery disease; three insulin-dependent diabetes mellitus; one hypothyroidism; one myocardial infarction; and one basilar artery stenosis). Five were on anticoagulant premedication (antiplatelet and oral anticoagulants). No patient had pre-diagnosed osteoporosis. 

### 3.2. Madeira Perspective

The ambulance service was alerted at 18:29 local time. The first ambulance, with an emergency medical technician, arrived at 18:38. Later, 3 physicians and 21 ambulances were transferred to the scene. Firefighters performed technical rescue. Transport time from the accident scene to the hospital was 10 min. In total, 28 patients were transported. Apart from four endotracheal airway securing measures, no invasive measures were carried out at the scene. At 18:31, a mass casualty pre-warning was given to the only available hospital, the Serviço Regional de Saúde da Região Autónoma da Madeira (SESARAM, EPRAM). The hospital was a Level III Trauma Center with 18 intensive care unit (ICU) beds, 2 trauma bays, and 11 operating rooms. 

All 28 prehospital patients were transported to this hospital. At 19:02, the first patient reached the emergency department. At 19:25, the first ICU admission took place and at 20:30, the first surgery. At 23:30, the last patient was admitted to the ward. In the first 24 h, 6 surgeries, 6 red blood cell transfusions, 23 computed tomography (CT) scans, and 30 radiographs were performed. Cross-sectional imaging was available in the emergency department. The following injury patterns were identified: -Orthopedic trauma to limbs and pelvis: multiple cases-Head trauma: one patient with severe traumatic brain injury died in the ICU, and another had severe craniofacial trauma-Thoracic trauma: two patients had pulmonary contusions and hemothorax-Abdominal trauma: one patient with traumatic duodenal laceration was laparatomized

The initial staffing of 18 physicians, including 9 surgeons and 4 anesthesiologists, and 21 nurses was augmented during the course by 9 surgeons, 20 nurses, and other personnel, such as psychologists, social workers, and administrative staff. The surgical specialties represented were abdominal, orthopedic trauma, cardiothoracic, neurological (head and spine), vascular, and plastics. The staff were trained in the standard trauma course formats like ATLS.

All 28 patients received a full initial assessment consisting of primary and secondary survey with adjuncts within the first 24 h after admission.

### 3.3. AE Perspective

Three days after the accident, patients were taken over at Madeira Airport by the crew of the A310-304 MRTT from the local ambulance service and prepared for air transport on the same day. Of the 15 patients, 1 was intubated and ventilated, while the remaining 14 were spontaneous breathing and conscious. Initially, median blood pressure was 136/75 mmHg, heart rate was 76 bpm (IQR 65–80), and peripheral oxygen saturation was 95.5% (IQR 92.0–98.8%). Six of the patients had a peripheral venous cannula (PVC, venous access), and one had a central venous catheter (CVC). Another had an arterial line for measuring blood pressure (AA). Three other patients received a PVC and one an AA in-flight. Seven patients needed supplemental oxygen during the flight since their spontaneous breathing was maintained, and one received non-invasive Continuous Positive Airway Pressure (CPAP) support. Another patient with a lung contusion had to be intubated and ventilated during the flight due to worsening pulmonary conditions and respiratory exhaustion. In Cologne, the median blood pressure was 135/80 mmHg and the heart rate was 80 bpm (IQR 70–85). There were no significant changes in vital signs during the flight (*p* > 0.05). The patients were under the care of the medical staff of the AE flight for a median of 5 h (min. 4 h, max. 7 h) from the takeover in Madeira until the handover to the local rescue service in Cologne (Figure 2).

### 3.4. Cologne Perspective

The receiving hospital received patient data (number, age, triage category, known injury pattern, and therapy given and ongoing) 24 h before AE. The team on call in the trauma department (one consultant, one fellow, and one resident) was augmented 2 h before the pre-planned admission time by the night shift team. Together with the hospital management team, seven trauma doctors were present (one director, two consultants, two fellows, and two residents). For each patient, the responsible physician, nurse, admission location in the emergency department, and ward room were defined pre-arrival. Patients of the red (1) and yellow (3) triage categories were treated in the emergency department with one responsible physician per category. Patients in the green category were treated in the consultation area of the traumatology clinic (one resident per six patients). This distribution was graphed with contact details and communicated in a briefing to the whole team 1.5 h before the first patient’s arrival (Figure 3). The director and consultants were not scheduled in patient care, but had coordinating tasks (one triage on arrival and one planning further diagnostics and therapy after admission). At no time was the hospital de-registered as a Level 1 trauma center for regular patients. 

The first patient arrived at 17:45 and the last patient at 18:50. The patients in the red and yellow categories arrived at the clinic between 18:00 and 18:40. The last patient was transferred from the admission rooms to the ward at 21:05. The patients were transferred to the wards only with a treatment plan approved by the consultant. 

The one patient intubated during flight was triaged in the red category on admission, being in the yellow category before. The median Injury Severity Score ISS was 10.0 (IQR 8.0–18.0), with 5 patients ISS > 16).

In the second hospital phase in Germany, in total 82 radiological images were prepared (56 X-rays, 26 CT scans). That adds up to around 5.5 images per patient transferred. Imaging can be divided into two phases. In the first 48 h—the “diagnostic” phase—57% of the CT scans (*n* = 15) and 48% of X-rays (*n* = 27) were taken in order to identify all relevant injuries. In the second phase—the re-assessment phase—imaging was used to control operative results and conservative healing processes (for example: 17 X-rays of the chest).

Additional CT diagnosis indicated the stabilization of unstable spine fractures (AO B1 fractures, “Chance-like”) in four patients. In three injuries, the primary assessment of fracture stability was revised, and one fracture was captured in the admission survey (the third survey following ATLS) for the first time. Osteosynthesis was also performed on a distal fibula (AO C2), and a soft tissue injury to the head was treated by plastic surgery using vacuum-assisted closure therapy and swing flaps. No emergency surgeries were performed on the day of admission. Surgical treatment was performed in the first four days post-admission (Figure 4).

Of the evaluated patients, 80% (8 out of 10) showed an increased risk for post-traumatic stress disorder (PTSD). Talk therapy based on the concept of somatic experience (SE) according to P. Levine was started in individual consultations with these patients. SE is reportedly suitable for the reduction in the development of PTSD after civilian trauma [7,8]. Every patient at risk received at least two talk therapy sessions (day 1 and 3 after admission). Three patients indicated the need for further sessions (day 5 after admission). Every patient refused offered oral psychopharmacological treatment.

Poor FSQ scores and the mental component of the SF-12 at the follow-up showed no significant correlation (*p* = 0.121).

Four patients directly admitted to the intensive care unit (ICU) stayed there for a median of 5 days (IQR 1–21). For all patients the hospital stay lasted for a median of 11 days (IQR 10–18). All but one patient were discharged home. One patient was transferred to a neurological rehabilitation institution.

The identification of one patient had to be revised since the language barrier and delirium probably caused an incorrect identification in Madeira.

### 3.5. Follow-Up Patients

Follow-up for 14 of 15 patients (93.3%) was carried out during a median follow-up period of 16 months (IQR 16–21). One patient died of a brain tumor unrelated to the accident. Compared with the German SF-12 reference, the median physical component score of the SF-12 was 49.7 (IQR 33.9–51.8, the median difference was −5.4 [IQR −17.1–0.33]), and the median mental component score was 50.7 (IQR 42.3–54.9), with a median difference of 0.7 (IQR −7.7–4.8). The patients expressed satisfaction with the treatment outcome. HRQoL was rated lower compared to pre-accident conditions, which served as reference in the questionnaires. No formal score data of the patients were available from the time before the accident. Of the patients, 53% (8/15) were still receiving medical treatment (six by physiotherapists and three by psychologists, one receiving both), and 47% (7/15) reported relevant accident-related limitations.

## 4. Discussion

The Advanced Trauma Life Support (ATLS) concept developed by the American College of Surgeons for the systematic, priority-oriented primary care of seriously injured patients includes repeated surveys of a patient at different times. In the primary survey, life-threatening injuries are to be identified and remedied (Madeira). The secondary survey, as a part of the initial assessment, should identify additional threats to life or limb and is used to identify all injuries sustained (Madeira). The third survey should monitor the course of injuries identified in the primary and secondary survey (Cologne) [9,10]. The resource-related transfer of patients should not be delayed by measures at any time. Following ATLS standards is able to reduce preventable mortality [11]. 

From the compilation of the different perspectives of care and the temporal processing of the individual injury histories, six relevant key findings emerged that should be used in future AE missions.

### 4.1. Madeira, Portugal

Prehospital and clinical care in Madeira showed a structured approach and clear communication. This was based on the high level of specific training, especially Advanced Trauma Life Support (ATLS) and Medical Response to Major Incidents and Disasters (MRMI) training, of the medical staff. 

As enough trained personnel were available quickly in the emergency room, performing the primary surveys was unproblematic and secondary surveys were partially started but not fully finished (missing cross-sectional imaging in stable patients). The early use of cross-sectional imaging, i.e., whole-body CTs, improves survival of blunt trauma patients when used in trauma resuscitation (number needed to scan: 17 to 32 patients) [12].

### 4.2. Cologne, Germany 

In Germany, ATLS was continued for every patient by a detailed tertiary survey. The task of this survey was to recognize transport-related deterioration of known injuries, to critically review treatment concepts, and to identify all unknown injuries.

In the injured collective, stability was reassessed in a relevant number of the spinal fractures (75%, 3/4) and one unstable spinal injury was newly identified. The other fractures had cross-sectional imaging before transfer. To avoid spinal cord injury, the correct assessment of the stability of a spinal fracture is of the highest priority, especially as the assessment has immediate implications for the transport section (en bloc positioning, supine position). It is therefore recommended to use international valid fracture classification systems (AOSpine thoracolumbar spine injury classification system), as these have a high interobserver reliability even among inexperienced users [13,14]. This is particularly required when advanced diagnostics for the final stability assessment (CT imaging) have not yet been completed, or digital imaging and communications of images are not available for assessment [15,16,17,18]. The classification provides modifiers (M1) for these unclear situations [14]. In case spinal stability cannot be assessed in the primary or secondary survey, patients should be treated as having unstable injuries. Therefore, they have spine immobilisation for transfer. Otherwise, there is an immanent risk of deterioration of the neurological outcome during transport since up to four transfers (bed to stretcher and back) are needed (**first key finding**).

Precise knowledge of the trauma mechanism (multiple rollovers of the vehicle with flexion-distraction and possible spinal rotation) and the individual risk of injury in the specific collective are also useful for the secondary care hospital [19].

The stresses of flight (hypoxia, gravitational forces, barometric pressure changes, thermal changes, vibration, humidity, noise, and fatigue) can worsen the clinical condition of a patient during airborne AE transport [20]. Therefore, invasive measures become necessary while the patient is still in-flight or directly upon admission to the destination hospital. In particular, elderly patients with pre-existing conditions and thoracic trauma have an increased risk of in-flight deterioration. Therefore, the ability of spontaneous breathing is critically reviewed and the AE team strive for proper monitoring of blood gases by arterial lines since AE mission are usually not conducted at sea-level cabin pressure. The supplementation of oxygen is generously used [21]. 

As a **second key finding**, it is recommended to maintain an ICU capacity (respirator) higher than announced and to have additional room options in the emergency department since deterioration of patients in-flight should be expected. 

Therefore, it is recommended to establish a secure communication channel with the destination hospital before the start of the transport that is maintained for the entire transport in order to immediately obtain notice of changing patient status and different resource demand (**third key finding**).

The patients showed a high risk for PTSD. Early exploration of the patients is recommended. Somatic experience seems to be a valuable tool of early intervention to avoid PTSD, at least until the time of follow-up. Inpatient psychotherapeutic support should be followed by outpatient psychotherapy in order to more effectively reduce symptoms 18 months after trauma [22]. At follow-up, four patients (50 percent) still needed outpatient psychotherapy. Trauma-focused therapies like cognitive processing therapy (CPT) and prolonged exposure retained diagnosis after treatment in up to two-thirds of patients, as shown in military-related PTSD [23]. 

The large number of patients usually exceeds the personnel resources of the receiving hospital, which is why the use of established military resources (PTSD teams) should be assessed. Usually, armed forces are well aware of PTSD risk and have qualified staff readily available spread across a country [24] (**fourth key finding**).

In order to avoid incorrect identification of patients, it is recommended for each patient to perform a new self-identification after admission to the destination hospital in the patient’s native language, if their mental orientation is present (**fifth key finding**). 

AE missions in the context of humanitarian aid missions or on behalf of the German government for the transport of German citizens receive relevant media attention and accompaniment. It is advisable to establish structures to manage external communication with relatives and the media. For undisturbed patient care, relatives and media representatives should be looked after in separate rooms without direct patient access by staff of the hospital [25] (**sixth key finding**).

### 4.3. Limitations

The results need to be interpreted with respect to the retrospective study design. Due to pandemic restrictions, a physical examination of participants was not possible, hence a self-reporting survey design.

## 5. Conclusions

Almost 2 years after the event, half of the patients still received some form of rehabilitation and almost 50% reported that their quality of life was impaired as a result of the accident. These consequences persist despite the consequent use of ATLS and the considerable use of resources in caring for the patients. 

## Figures and Tables

**Figure 1 jcm-12-04556-f001:**
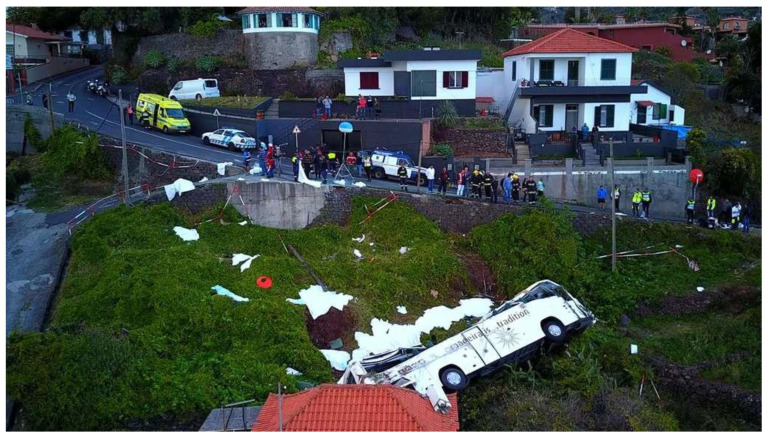
Aerial view of the accident site (©Getty Images).

**Figure 2 jcm-12-04556-f002:**
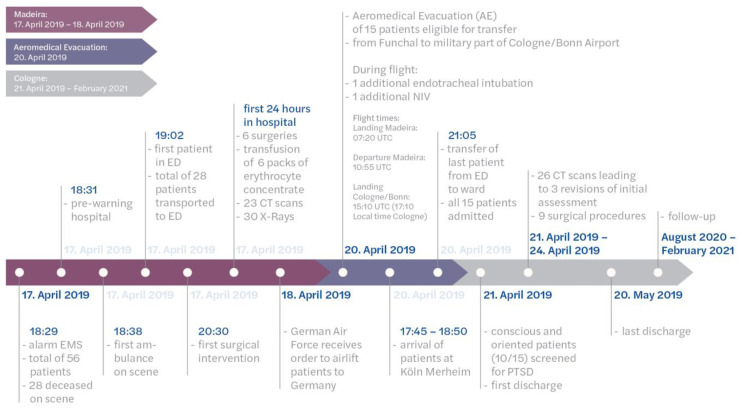
Schematic timeline of actions.

**Figure 3 jcm-12-04556-f003:**
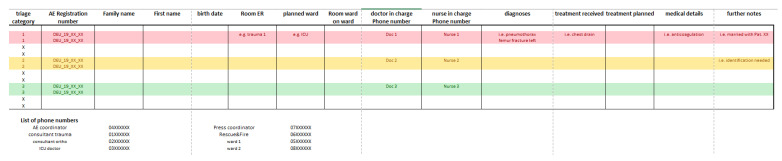
Sample document of patient documentation and workflow in Cologne.

**Figure 4 jcm-12-04556-f004:**
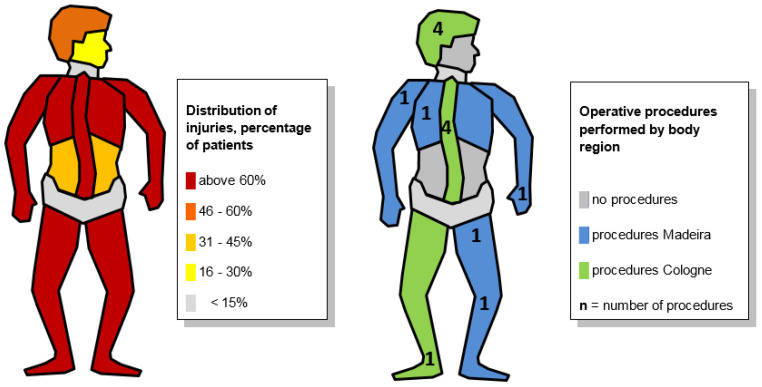
Chart of injuries and operative procedures by body region.

## Data Availability

Data available on request due to restrictions, e.g., privacy or ethical.

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
