# Peer review of "Performing Advanced Trauma Life Support (ATLS) across Borders: Midterm Follow-Up of the Aeromedical Evacuation after Civilian Bus Accident at Madeira"

_jcm, 2023, doi:10.3390/jcm12144556_

Round 1

Reviewer 1 Report

Refer to the attachment 

The writing structure should be improve (for example, 1 sentence per paragraph might not be an appropriate)

Author Response

Dear reviewer,

first, we would like to thank you for investing your valuable time in our article. Your comments are exceptionally constructive and fulley eligible. We therefore took intensive efforts to meet your expectations. Following are the responses to all your observations in a point-by-point response. All the suggestions and recommendations were accepted, and the changes are available in the revised manuscript. Please see the changes explained in the following paragraphs and in the revised manuscript.  You find all changes in the manuscript highlighted in blue color.

Lines 23-24: in complete sentences, what is available? ..... In Germany all available were reviewed for 23 data collection. Indeed there were words missing - all data sheets were assessed in Germany, we apologise! (line 29)
Lines 30-31: spell out the number at the beginning of the sentence (i.e., Eighty percent) or reword.
Changed (line 35)
Lines 52-57 and 66-68: writing styles, only a sentence for a paragraph?
Adapted
Line 61: reference start with 17?
We re-worked the entire reference section and added more literature. We do not understand why the list started with 17 - now it is 1.
Lines 78-81: where is the reference for this?
we added a reference´which is unfortunatly in German but contains a figure with all data taken.
Lines 81-84: error?
We commented on error correction. Basically we just double-checked data and took missing values from the paper based documentation.
Lines 95-98: describe the PTSD symptoms score and cut-off. This can explain the 8 out of 10 increased PTSD risk.
As adviced we added some literture with cut-off values.
Lines 234-236: Describe these scores in the method section (for example increase or decrease score indicating higher/lower symptoms/QoL.....)
We added an extensive part in the method section with literature (lines 103-111)
Line 238: all but one patient discharged home at 5 days or 11 days or early? Pls clarify
- hopefully we successfully clarified this part. Four patients who we were admissioned at ICU, stayed there for 5 days in median, hospital stay was 11 days for all patients in median
Lines 249-250: Does the HRQoL questionnaire was conducted before the accident?
there were no HRQoL data before accident, the patients were asked to compare their post-traumatic status with their well-being before the accident
Lines 306-306: increased risk of what?
We commented on the risk of in-flight deterioriation
Lines 311-313: Maybe explained a little more about why communication is important here. Somehow, its disconnected between findings 2 and 3.
We have re-written this paragraph. Stable communication enables constant update of patient status which we were lacking. We finally recieved more ICU patients than expected.
Discussion: Key findings should be discussed in more detail with citations. why it is important, and what has been done (A lot of research has been done for PTSD risk after physical trauma). We followed your advice by adding more literature and further describing your best-practice solutions. In the frist version of the manuscript we tried to keep the script very sleek.

We now hope to have settled all deficits of our study to our satisfaction.

Yours sincerely

Sebastian Imach, speaking for all authors of the paper

Reviewer 2 Report

The article is well written and well documented and I have the following positive considerations:

 Comprehensive description of the Incident: the article provides a detailed account of the bus crash incident in Caniço, Madeira, including the specific circumstances, causes, and outcomes. This level of information is essential for trauma surgeons and healthcare providers to understand the context and dynamics of such events. This enabled them to develop effective strategies and protocols.

Emphasis on ATLS-Based Management: The study highlights the application of Advanced Trauma Life Support (ATLS) principles in the management of injured patients. ATLS is a widely recognized and respected approach to trauma care. Its successful implementation at a mass casualty event provides reassurance and guidance for trauma surgeons and healthcare providers facing similar circumstances.

Real-World Application: The case study presented in the article reflects a real-world scenario and the challenges faced by trauma surgeons and healthcare providers. This practical approach enhances the applicability of the findings and recommendations, making them more useful and actionable for professionals working in similar settings. In In summary, this article provides trauma surgeons and healthcare providers dealing with mass casualties and emergencies with valuable insights into the medical and logistical challenges associated with such incidents. The comprehensive description of the incident, emphasis on ATLS-based management, and consideration of long-term outcomes contribute to the relevance and usefulness of this paper for professionals in the field.

I have six questions:

1. Incomplete secondary surveys in Madeira: The secondary surveys conducted in Madeira were partially started but not fully finished, with missing cross-sectional imaging in stable patients. This could have limited the identification of some significant injuries sustained by the patients. Can you comment on this?

2. It seems that the assessment of spinal fracture stability was not fully completed before transfer. A relevant number of spinal fractures required reassessment in Cologne. Can you comment on and discuss how an inaccurate stability assessment of spinal injury can lead to further complications during transport?

3. The stresses of flight can worsen the clinical condition of some patients, especially elderly patients with pre-existing conditions and thoracic trauma. Can you tell us if specific concerns were considered for some patients during airborne AE transport?

4. PTSD risk management: The study identified a high risk of post-traumatic stress disorder (PTSD) among patients. For early intervention, somatic experience was recommended as a valuable tool for exploring patients. However, due to the large number of patients in such an emergency, personnel resources might be insufficient. The utilization of established military resources, such as PTSD teams, should be considered. Can you comment on this issue? These critiques and considerations provide a basis for further discussion and improvement in future AE missions. This is done by taking into account the limitations of the retrospective study design and the need for self-reporting surveys

5. Line 170: PVK is an acronym for what?

6. Please briefly describe the German SF-12 score: is it a questionnaire for physical and mental health?

Author Response

Dear reviewer,

first, we would like to thank you for investing your valuable time in our article. Your comments are exceptionally benevolent, constructive and eligible. We therefore took intensive efforts to meet your expectations. Following are the responses to all your observations in a point-by-point response. All the suggestions and recommendations were accepted, and the changes are available in the revised manuscript. Please see the changes explained in the following paragraphs and in the revised manuscript.  You find all changes in the manuscript highlighted in blue color.

  1. Incomplete secondary surveys in Madeira: The secondary surveys conducted in Madeira were partially started but not fully finished, with missing cross-sectional imaging in stable patients. This could have limited the identification of some significant injuries sustained by the patients. Can you comment on this? We added the valuable work of Huber-Wagner et al. which demonstrated a survival benefit of patients with blunt trauma recieving CT scans (lines 271-273, citation 12)
  2. It seems that the assessment of spinal fracture stability was not fully completed before transfer. A relevant number of spinal fractures required reassessment in Cologne. Can you comment on and discuss how an inaccurate stability assessment of spinal injury can lead to further complications during transport? We added a comment addressing the relevant number of patient transfers during AE missions (lines 291-293)
  3. The stresses of flight can worsen the clinical condition of some patients, especially elderly patients with pre-existing conditions and thoracic trauma. Can you tell us if specific concerns were considered for some patients during airborne AE transport? We picked up the measures taken bei the AE crew in the discussion section addressing non sea level cabin pressure (oxygen etc., lines 301-305)
  4. PTSD risk management: The study identified a high risk of post-traumatic stress disorder (PTSD) among patients. For early intervention, somatic experience was recommended as a valuable tool for exploring patients. However, due to the large number of patients in such an emergency, personnel resources might be insufficient. The utilization of established military resources, such as PTSD teams, should be considered. Can you comment on this issue? These critiques and considerations provide a basis for further discussion and improvement in future AE missions. This is done by taking into account the limitations of the retrospective study design and the need for self-reporting surveys. We elaborated this finding with more literature and highlighted our approach compared to literature findings. In our experience armed forces still have those teams in place while we had to organise them first. (lines 313-324)
  5. Line 170: PVK is an acronym for what? We are sorry, this has been a typo error. PVC as venous access was meant (line166)
  6. Please briefly describe the German SF-12 score: is it a questionnaire for physical and mental health? We added a section in methods with literature (lines 103-111)

We now hope to have settled all deficits of our study to our satisfaction.

Yours sincerely

Sebastian Imach, speaking for all authors of the paper

Round 2

Reviewer 1 Report

Thanks for your revised manuscript. 

The copy-editing of your manuscript is required.